# Learning Data Manipulation for Augmentation and Weighting

**Zhiting Hu**[1,2]*, **Bowen Tan**[1]*, **Ruslan Salakhutdinov**[1], **Tom Mitchell**[1], **Eric P. Xing**[1,2]
[1]Carnegie Mellon University, [2]Petuum Inc.
{zhitingh,btan2,rsalakhu,tom.mitchell}@cs.cmu.edu, eric.xing@petuum.com

## Abstract

Manipulating data, such as weighting data examples or augmenting with new instances, has been increasingly used to improve model training. Previous work has studied various rule- or learning-based approaches designed for specific types of data manipulation. In this work, we propose a new method that supports learning different manipulation schemes with the same gradient-based algorithm. Our approach builds upon a recent connection of supervised learning and reinforcement learning (RL), and adapts an off-the-shelf reward learning algorithm from RL for joint data manipulation learning and model training. Different parameterization of the "data reward" function instantiates different manipulation schemes. We showcase data augmentation that learns a text transformation network, and data weighting that dynamically adapts the data sample importance. Experiments show the resulting algorithms significantly improve the image and text classification performance in low data regime and class-imbalance problems.

## 1   Introduction

The performance of machines often crucially depend on the amount and quality of the data used for training. It has become increasingly ubiquitous to *manipulate* data to improve learning, especially in low data regime or in presence of low-quality datasets (e.g., imbalanced labels). For example, *data augmentation* applies label-preserving transformations on original data points to expand the data size; *data weighting* assigns an importance weight to each instance to adapt its effect on learning; and *data synthesis* generates entire artificial examples. Different types of manipulation can be suitable for different application settings.

Common data manipulation methods are usually designed manually, e.g., augmenting by flipping an image or replacing a word with synonyms, and weighting with inverse class frequency or loss values [10, 32]. Recent work has studied automated approaches, such as learning the composition of augmentation operators with reinforcement learning [38, 5], deriving sample weights adaptively from a validation set via meta learning [39], or learning a weighting network by inducing a curriculum [21]. These learning-based approaches have alleviated the engineering burden and produced impressive results. However, the algorithms are usually designed specifically for certain types of manipulation (e.g., either augmentation or weighting) and thus have limited application scope in practice.

In this work, we propose a new approach that enables learning for different manipulation schemes with the same single algorithm. Our approach draws inspiration from the recent work [46] that shows equivalence between the data in supervised learning and the reward function in reinforcement learning. We thus adapt an off-the-shelf *reward learning* algorithm [52] to the supervised setting for automated data manipulation. The marriage of the two paradigms results in a simple yet general algorithm, where various manipulation schemes are reduced to different parameterization of the

---

*data reward*. Free parameters of manipulation are learned jointly with the target model through efficient gradient descent on validation examples. We demonstrate instantiations of the approach for automatically fine-tuning an augmentation network and learning data weights, respectively.

We conduct extensive experiments on text and image classification in challenging situations of very limited data and imbalanced labels. Both augmentation and weighting by our approach significantly improve over strong base models, even though the models are initialized with large-scale pretrained networks such as BERT [7] for text and ResNet [14] for images. Our approach, besides its generality, also outperforms a variety of dedicated rule- and learning-based methods for either augmentation or weighting, respectively. Lastly, we observe that the two types of manipulation tend to excel in different contexts: augmentation shows superiority over weighting with a small amount of data available, while weighting is better at addressing class imbalance problems.

The way we derive the manipulation algorithm represents a general means of problem solving through algorithm extrapolation between learning paradigms, which we discuss more in section 6.

## 2    Related Work

Rich types of data manipulation have been increasingly used in modern machine learning pipelines. Previous work each has typically focused on a particular manipulation type. Data augmentation that perturbs examples without changing the labels is widely used especially in vision [44, 26] and speech [24, 36] domains. Common heuristic-based methods on images include cropping, mirroring, rotation [26], and so forth. Recent work has developed automated augmentation approaches [5, 38, 28, 37, 47]. Xie et al. [50] additionally use large-scale unlabeled data. Cubuk et al. [5], Ratner et al. [38] learn to induce the composition of data transformation operators. Instead of treating data augmentation as a policy in reinforcement learning [5], we formulate manipulation as a reward function and use efficient stochastic gradient descent to learn the manipulation parameters. Text data augmentation has also achieved impressive success, such as contextual augmentation [25, 49], back-translation [42], and manual approaches [48, 2]. In addition to perturbing the input text as in classification tasks, text generation problems expose opportunities to adding noise also in the output text, such as [35, 51]. Recent work [46] shows output nosing in sequence generation can be treated as an intermediate approach in between supervised learning and reinforcement learning, and developed a new sequence learning algorithm that interpolates between the spectrum of existing algorithms. We instantiate our approach for text contextual augmentation as in [25, 49], but enhance the previous work by additionally fine-tuning the augmentation network jointly with the target model.

Data weighting has been used in various algorithms, such as AdaBoost [10], self-paced learning [27], hard-example mining [43], and others [4, 22]. These algorithms largely define sample weights based on training loss. Recent work [21, 8] learns a separate network to predict sample weights. Of particular relevance to our work is [39] which induces sample weights using a validation set. The data weighting mechanism instantiated by our framework has a key difference in that samples weights are treated as parameters that are updated iteratively, instead of re-estimated from scratch at each step. We show improved performance of our approach. Besides, our data manipulation approach is derived based on a different perspective of reward learning, instead of meta-learning as in [39].

Another popular type of data manipulation involves data synthesis, which creates entire artificial samples from scratch. GAN-based approaches have achieved impressive results for synthesizing conditional image data [3, 34]. In the text domain, controllable text generation [17] presents a way of co-training the data generator and classifier in a cyclic manner within a joint VAE [23] and wake-sleep [15] framework. It is interesting to explore the instantiation of the present approach for adaptive data synthesis in the future.

## 3    Background

We first present the relevant work upon which our automated data manipulation is built. This section also establishes the notations used throughout the paper.

Let $x$ denote the input and $y$ the output. For example, in text classification, $x$ can be a sentence and $y$ is the sentence label. Denote the model of interest as $p_\theta(y|x)$, where $\theta$ is the model parameters to be

learned. In supervised setting, given a set of training examples $\mathcal{D} = \{(\boldsymbol{x}^*, y^*)\}$, we learn the model by maximizing the data log-likelihood.

**Equivalence between Data and Reward**   The recent work [46] introduced a unifying perspective of reformulating maximum likelihood supervised learning as a special instance of a policy optimization framework. In this perspective, data examples providing supervision signals are equivalent to a specialized reward function. Since the original framework [46] was derived for sequence generation problems, here we present a slightly adapted formulation for our context of data manipulation.

To connect the maximum likelihood supervised learning with policy optimization, consider the model $p_\theta(y|\boldsymbol{x})$ as a policy that takes "action" $y$ given the "state" $\boldsymbol{x}$. Let $R(\boldsymbol{x}, y|\mathcal{D}) \in \mathbb{R}$ denote a reward function, and $p(\boldsymbol{x})$ be the empirical data distribution which is known given $\mathcal{D}$. Further assume a variational distribution $q(\boldsymbol{x}, y)$ that factorizes as $q(\boldsymbol{x}, y) = p(\boldsymbol{x})q(y|\boldsymbol{x})$. A variational policy optimization objective is then written as:

$$\mathcal{L}(q, \boldsymbol{\theta}) = \mathbb{E}_{q(\boldsymbol{x},y)}\left[R(\boldsymbol{x}, y|\mathcal{D})\right] - \alpha\text{KL}\big(q(\boldsymbol{x}, y)\|p(\boldsymbol{x})p_\theta(y|\boldsymbol{x})\big) + \beta\text{H}(q), \tag{1}$$

where $\text{KL}(\cdot\|\cdot)$ is the Kullback–Leibler divergence; $\text{H}(\cdot)$ is the Shannon entropy; and $\alpha, \beta > 0$ are balancing weights. The objective is in the same form with the RL-as-inference formalism of policy optimization [e.g., 6, 29, 1]. Intuitively, the objective maximizes the expected reward under $q$, and enforces the model $p_\theta$ to stay close to $q$, with a maximum entropy regularization over $q$. The problem is solved with an EM procedure that optimizes $q$ and $\boldsymbol{\theta}$ alternatingly:

$$\begin{aligned} \text{E-step:} \quad & q'(\boldsymbol{x}, y) = \exp\left\{\frac{\alpha\log p(\boldsymbol{x})p_\theta(y|\boldsymbol{x}) + R(\boldsymbol{x}, y|\mathcal{D})}{\alpha + \beta}\right\} / Z, \\ \text{M-step:} \quad & \boldsymbol{\theta}' = \arg\max_\theta \mathbb{E}_{q'(\boldsymbol{x},y)}\left[\log p_\theta(y|\boldsymbol{x})\right], \end{aligned} \tag{2}$$

where $Z$ is the normalization term. With the established framework, it is easy to show that the above optimization procedure reduces to maximum likelihood learning by taking $\alpha \to 0, \beta = 1$, and the reward function:

$$R_\delta(\boldsymbol{x}, y|\mathcal{D}) = \left\{ \begin{array}{ll} 1 & \text{if } (\boldsymbol{x}, y) \in \mathcal{D} \\ -\infty & \text{otherwise.} \end{array} \right. \tag{3}$$

That is, a sample $(\boldsymbol{x}, y)$ receives a unit reward only when it matches a training example in the dataset, while the reward is negative infinite in all other cases. To make the equivalence to maximum likelihood learning clearer, note that the above M-step now reduces to

$$\boldsymbol{\theta}' = \arg\max_\theta \mathbb{E}_{p(\boldsymbol{x})\exp\{R_\delta\}/Z}\left[\log p_\theta(y|\boldsymbol{x})\right], \tag{4}$$

where the joint distribution $p(\boldsymbol{x})\exp\{R_\delta\}/Z$ equals the empirical data distribution, which means the M-step is in fact maximizing the data log-likelihood of the model $p_\theta$.

**Gradient-based Reward Learning**   There is a rich line of research on learning the reward in reinforcement learning. Of particular interest to this work is [52] which learns a parametric *intrinsic* reward that additively transforms the original task reward (a.k.a *extrinsic* reward) to improve the policy optimization. For consistency of notations with above, formally, let $p_\theta(y|\boldsymbol{x})$ be a policy where $y$ is an action and $\boldsymbol{x}$ is a state. Let $R_\phi^{in}$ be the intrinsic reward with parameters $\phi$. In each iteration, the policy parameter $\boldsymbol{\theta}$ is updated to maximize the joint rewards, through:

$$\boldsymbol{\theta}' = \boldsymbol{\theta} + \gamma\nabla_\theta\mathcal{L}^{ex+in}(\boldsymbol{\theta}, \phi), \tag{5}$$

where $\mathcal{L}^{ex+in}$ is the expectation of the sum of extrinsic and intrinsic rewards; and $\gamma$ is the step size. The equation shows $\boldsymbol{\theta}'$ depends on $\phi$, thus we can write as $\boldsymbol{\theta}' = \boldsymbol{\theta}'(\phi)$.

The next step is to optimize the intrinsic reward parameters $\phi$. Recall that the *ultimate measure* of the performance of a policy is the value of extrinsic reward it achieves. Therefore, a *good* intrinsic reward is supposed to, when the policy is trained with it, increase the eventual extrinsic reward. The update to $\phi$ is then written as:

$$\phi' = \phi + \gamma\nabla_\phi\mathcal{L}^{ex}(\boldsymbol{\theta}'(\phi)). \tag{6}$$

That is, we want the expected extrinsic reward $\mathcal{L}^{ex}(\boldsymbol{\theta}')$ of the new policy $\boldsymbol{\theta}'$ to be maximized. Since $\boldsymbol{\theta}'$ is a function of $\phi$, we can directly backpropagate the gradient through $\boldsymbol{\theta}'$ to $\phi$.

**Algorithm 1** Joint Learning of Model and Data Manipulation

---

**Input:** The target model $p_\theta(y|\boldsymbol{x})$
        The data manipulation function $R_\phi(\boldsymbol{x}, y|\mathcal{D})$
        Training set $\mathcal{D}$, validation set $\mathcal{D}^v$
1: Initialize model parameter $\theta$ and manipulation parameter $\phi$
2: **repeat**
3:     Optimize $\theta$ on $\mathcal{D}$ enriched with data manipulation
      through Eq.(7)
4:     Optimize $\phi$ by maximizing data log-likelihood on $\mathcal{D}^v$
      through Eq.(8)
5: **until** convergence
**Output:** Learned model $p_{\theta^*}(y|\boldsymbol{x})$ and manipulation $R_{\phi^*}(y, \boldsymbol{x}|\mathcal{D})$

---

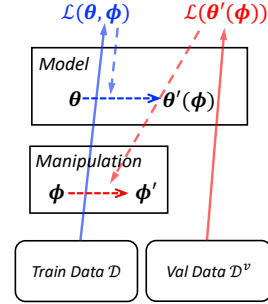

**Figure 1:** Algorithm Computation. Blue arrows denote learning model $\theta$. Red arrows denote learning manipulation $\phi$. Solid arrows denote forward pass. Dashed arrows denote backward pass and parameter updates.

# 4 Learning Data Manipulation

## 4.1 Method

**Parameterizing Data Manipulation** We now develop our approach of learning data manipulation, through a novel marriage of supervised learning and the above reward learning. Specifically, from the policy optimization perspective, due to the $\delta$-function reward (Eq.3), the standard maximum likelihood learning is restricted to use only the exact training examples $\mathcal{D}$ in a uniform way. A natural idea of enabling data manipulation is to relax the strong restrictions of the $\delta$-function reward and instead use a relaxed reward $R_\phi(\boldsymbol{x}, y|\mathcal{D})$ with parameters $\phi$. The relaxed reward can be parameterized in various ways, resulting in different types of manipulation. For example, when a sample $(\boldsymbol{x}, y)$ matches a data instance, instead of returning constant 1 by $R_\delta$, the new $R_\phi$ can return varying reward values depending on the matched instance, resulting in a data weighting scheme. Alternatively, $R_\phi$ can return a valid reward even when $\boldsymbol{x}$ matches a data example only in part, or $(\boldsymbol{x}, y)$ is an entire new sample not in $\mathcal{D}$, which in effect makes data augmentation and data synthesis, respectively, in which cases $\phi$ is either a data transformer or a generator. In the next section, we demonstrate two particular parameterizations for data augmentation and weighting, respectively.

We thus have shown that the diverse types of manipulation all boil down to a parameterized data reward $R_\phi$. Such an concise, uniform formulation of data manipulation has the advantage that, once we devise a method of learning the manipulation parameters $\phi$, the resulting algorithm can directly be applied to automate any manipulation type. We present a learning algorithm next.

**Learning Manipulation Parameters** To learn the parameters $\phi$ in the manipulation reward $R_\phi(\boldsymbol{x}, y|\mathcal{D})$, we could in principle adopt any off-the-shelf reward learning algorithm in the literature. In this work, we draw inspiration from the above gradient-based reward learning (section 3) due to its simplicity and efficiency. Briefly, the objective of $\phi$ is to maximize the *ultimate measure* of the performance of model $p_\theta(\boldsymbol{y}|\boldsymbol{x})$, which, in the context of supervised learning, is the model performance on a held-out validation set.

The algorithm optimizes $\theta$ and $\phi$ alternatingly, corresponding to Eq.(5) and Eq.(6), respectively. More concretely, in each iteration, we first update the model parameters $\theta$ in analogue to Eq.(5) which optimizes intrinsic reward-enriched objective. Here, we optimize the log-likelihood of the training set enriched with data manipulation. That is, we replace $R_\delta$ with $R_\phi$ in Eq.(4), and obtain the augmented M-step:

$$\boldsymbol{\theta}' = \arg\max_\theta \mathbb{E}_{p(\boldsymbol{x})\exp\{R_\phi(\boldsymbol{x},y|\mathcal{D})\}/Z}\big[\log p_\theta(y|\boldsymbol{x})\big]. \tag{7}$$

By noticing that the new $\boldsymbol{\theta}'$ depends on $\phi$, we can write $\boldsymbol{\theta}'$ as a function of $\phi$, namely, $\boldsymbol{\theta}' = \boldsymbol{\theta}'(\phi)$. The practical implementation of the above update depends on the actual parameterization of manipulation $R_\phi$, which we discuss in more details in the next section.

The next step is to optimize $\phi$ in terms of the model validation performance, in analogue to Eq.(6). Formally, let $\mathcal{D}^v$ be the validation set of data examples. The update is then:

$$
\begin{aligned}
\phi' &= \arg\max_{\phi} \mathbb{E}_{p(\boldsymbol{x}) \exp\{R_\delta(\boldsymbol{x},y|\mathcal{D}^v)\}/Z} \big[ \log p_{\theta'}(y|\boldsymbol{x}) \big] \\
&= \arg\max_{\phi} \mathbb{E}_{(\boldsymbol{x},y)\sim\mathcal{D}^v} \big[ \log p_{\theta'}(y|\boldsymbol{x}) \big],
\end{aligned}
\tag{8}
$$

where, since $\boldsymbol{\theta}'$ is a function of $\phi$, the gradient is backpropagated to $\phi$ through $\boldsymbol{\theta}'(\phi)$. Taking data weighting for example where $\phi$ is the training sample weights (more details in section 4.2), the update is to optimize the weights of training samples so that the model performs best on the validation set.

The resulting algorithm is summarized in Algorithm 1. Figure 1 illustrates the computation flow. Learning the manipulation parameters effectively uses a held-out validation set. We show in our experiments that a very small set of validation examples (e.g., 2 labels per class) is enough to significantly improve the model performance in low data regime.

It is worth noting that some previous work has also leveraged validation examples, such as learning data augmentation with policy gradient [5] or inducing data weights with meta-learning [39]. Our approach is inspired from a distinct paradigm of (intrinsic) reward learning. In contrast to [5] that treats data augmentation as a policy, we instead formulate manipulation as a reward function and enable efficient stochastic gradient updates. Our approach is also more broadly applicable to diverse data manipulation types than [39, 5].

## 4.2 Instantiations: Augmentation & Weighting

As a case study, we show two parameterizations of $R_\phi$ which instantiate distinct data manipulation schemes. The first example learns augmentation for text data, a domain that has been less studied in the literature compared to vision and speech [25, 12]. The second instance focuses on automated data weighting, which is applicable to any data domains.

**Fine-tuning Text Augmentation**

The recent work [25, 49] developed a novel contextual augmentation approach for text data, in which a powerful pretrained language model (LM), such as BERT [7], is used to generate substitutions of words in a sentence. Specifically, given an observed sentence $\boldsymbol{x}^*$, the method first randomly masks out a few words. The masked sentence is then fed to BERT which fills the masked positions with new words. To preserve the original sentence class, the BERT LM is retrofitted as a label-conditional model, and trained on the task training examples. The resulting model is then fixed and used to augment data during the training of target model. We denote the augmentation distribution as $g_{\phi_0}(\boldsymbol{x}|\boldsymbol{x}^*, \boldsymbol{y}^*)$, where $\phi_0$ is the fixed BERT LM parameters.

The above process has two drawbacks. First, the LM is fixed after fitting to the task data. In the subsequent phase of training the target model, the LM augments data without knowing the state of the target model, which can lead to sub-optimal results. Second, in the cases where the task dataset is small, the LM can be insufficiently trained for preserving the labels faithfully, resulting in noisy augmented samples.

To address the difficulties, it is beneficial to apply the proposed learning data manipulation algorithm to additionally fine-tune the LM *jointly* with target model training. As discussed in section 4, this reduces to properly parameterizing the data reward function:

$$
R_\phi^{aug}(\boldsymbol{x}, y|\mathcal{D}) = \begin{cases} 1 & \text{if } \boldsymbol{x} \sim g_\phi(\boldsymbol{x}|\boldsymbol{x}^*, y), \ (\boldsymbol{x}^*, y) \in \mathcal{D} \\ -\infty & \text{otherwise.} \end{cases}
\tag{9}
$$

That is, a sample $(\boldsymbol{x}, y)$ receives a unit reward when $y$ is the true label and $\boldsymbol{x}$ is the augmented sample by the LM (instead of the exact original data $\boldsymbol{x}^*$). Plugging the reward into Eq.(7), we obtain the data-augmented update for the model parameters:

$$
\boldsymbol{\theta}' = \arg\max_{\theta} \mathbb{E}_{\boldsymbol{x}\sim g_\phi(\boldsymbol{x}|\boldsymbol{x}^*, y), \ (\boldsymbol{x}^*, y)\sim\mathcal{D}} \big[ \log p_\theta(y|\boldsymbol{x}) \big].
\tag{10}
$$

That is, we pick an example from the training set, and use the LM to create augmented samples, which are then used to update the target model. Regarding the update of augmentation parameters $\phi$ (Eq.8), since text samples are discrete, to enable efficient gradient propagation through $\boldsymbol{\theta}'$ to $\phi$, we use a gumbel-softmax approximation [20] to $\boldsymbol{x}$ when sampling substitution words from the LM.

**Learning Data Weights**
We now demonstrate the instantiation of data weighting. We aim to assign an importance weight to each training example to adapt its effect on model training. We automate the process by learning the data weights. This is achieved by parameterizing $R_\phi$ as:

$$R_\phi^w(\boldsymbol{x}, y|\mathcal{D}) = \begin{cases} \phi_i & \text{if } (\boldsymbol{x}, y) = (\boldsymbol{x}_i^*, y_i^*), \ (\boldsymbol{x}_i^*, y_i^*) \in \mathcal{D} \\ -\infty & \text{otherwise,} \end{cases} \tag{11}$$

where $\phi_i \in \mathbb{R}$ is the weight associated with the $i$th example. Plugging $R_\phi^w$ into Eq.(7), we obtain the weighted update for the model $\boldsymbol{\theta}$:

$$\begin{aligned} \boldsymbol{\theta}' &= \arg\max_\theta \mathbb{E}_{(\boldsymbol{x}_i^*, y_i^*) \in \mathcal{D}, \ i \sim \text{softmax}(\phi_i)} \big[ \log p_\theta(y_i^*|\boldsymbol{x}_i^*) \big] \\ &= \arg\max_\theta \mathbb{E}_{(\boldsymbol{x}_i^*, y_i^*) \sim \mathcal{D}} \big[ \text{softmax}(\phi_i) \log p_\theta(y_i^*|\boldsymbol{x}_i^*) \big] \end{aligned} \tag{12}$$

In practice, when minibatch stochastic optimization is used, we approximate the weighted sampling by taking the softmax over the weights of only the minibatch examples. The data weights $\phi$ are updated with Eq.(8). It is worth noting that the previous work [39] similarly derives data weights based on their gradient directions on a validation set. Our algorithm differs in that the data weights are parameters maintained and updated throughout the training, instead of re-estimated from scratch in each iteration. Experiments show the parametric treatment achieves superior performance in various settings. There are alternative parameterizations of $R_\phi$ other than Eq.(11). For example, replacing $\phi_i$ in Eq.(11) with $\log \phi_i$ in effect changes the softmax normalization in Eq.(12) to linear normalization, which is used in [39].

## 5 Experiments

We empirically validate the proposed data manipulation approach through extensive experiments on learning augmentation and weighting. We study both text and image classification, in two difficult settings of low data regime and imbalanced labels[1].

### 5.1 Experimental Setup

**Base Models.** We choose strong pretrained networks as our base models for both text and image classification. Specifically, on text data, we use the BERT (base, uncased) model [7]; while on image data, we use ResNet-34 [14] pretrained on ImageNet. We show that, even with the large-scale pretraining, data manipulation can still be very helpful to boost the model performance on downstream tasks. Since our approach uses validation sets for manipulation parameter learning, for a fair comparison with the base model, we train the base model in two ways. The first is to train the model on the training sets as usual and select the best step using the validation sets; the second is to train on the merged training and validation sets for a fixed number of steps. The step number is set to the average number of steps selected in the first method. We report the results of both methods.

**Comparison Methods.** We compare our approach with a variety of previous methods that were designed for specific manipulation schemes: (1) For text data augmentation, we compare with the latest model-based augmentation [49] which uses a **fixed conditional BERT** language model for word substitution (section 4.2). As with base models, we also tried fitting the augmentatin model to both the training data and the joint training-validation data, and did not observe significant difference. Following [49], we also study a conventional approach that replaces words with their **synonyms** using WordNet [33]. (2) For data weighting, we compare with the state-of-the-art approach [39] that dynamically re-estimates sample weights in each iteration based on the validation set gradient directions. We follow [39] and also evaluate the commonly-used **proportion** method that weights data by inverse class frequency.

**Training.** For both the BERT classifier and the augmentation model (which is also based on BERT), we use Adam optimization with an initial learning rate of 4e-5. For ResNets, we use SGD optimization with a learning rate of 1e-3. For text data augmentation, we augment each minibatch by generating two or three samples for each data points (each with 1, 2 or 3 substitutions), and use both the samples and the original data to train the model. For data weighting, to avoid exploding value, we update the weight of each data point in a minibatch by decaying the previous weight value

| Model | SST-5 (40+2) | IMDB (40+5) | TREC (40+5) |
|---|---|---|---|
| Base model: BERT [7] | $33.32 \pm 4.04$ | $63.55 \pm 5.35$ | $88.25 \pm 2.81$ |
| Base model + val-data | $35.86 \pm 3.03$ | $63.65 \pm 3.32$ | $88.42 \pm 4.90$ |
| **Augment** Synonym | $32.45 \pm 4.59$ | $62.68 \pm 3.94$ | $88.26 \pm 2.76$ |
| Fixed augmentation [49] | $34.84 \pm 2.76$ | $63.65 \pm 3.21$ | $88.28 \pm 4.50$ |
| **Ours**: Fine-tuned augmentation | $\mathbf{37.03 \pm 2.05}$ | $\mathbf{65.62 \pm 3.32}$ | $\mathbf{89.15 \pm 2.41}$ |
| **Weight** Ren et al. [39] | $36.09 \pm 2.26$ | $63.01 \pm 3.33$ | $88.60 \pm 2.85$ |
| **Ours** | $\mathbf{36.51 \pm 2.54}$ | $\mathbf{64.78 \pm 2.72}$ | $\mathbf{89.01 \pm 2.39}$ |

**Table 1:** Accuracy of Data Manipulation on Text Classification. All results are averaged over 15 runs $\pm$ one standard deviation. The numbers in parentheses next to the dataset names indicate the size of the datasets. For example, (40+2) denotes 40 training instances and 2 validation instances *per class*.

**Figure 2:** Words predicted with the highest probabilities by the augmentation LM. Two tokens "striking" and "grey" are masked for substitution. The boxes in respective colors list the predicted words after training epoch 1 and 3, respectively. E.g., "stunning" is the most probable substitution for "striking" in epoch 1.

| Model | Pretrained | Not-Pretrained |
|---|---|---|
| Base model: ResNet-34 | $37.69 \pm 3.03$ | $22.98 \pm 2.81$ |
| Base model + val-data | $38.09 \pm 1.87$ | $23.42 \pm 1.47$ |
| Ren et al. [39] | $38.02 \pm 2.14$ | $23.44 \pm 1.63$ |
| **Ours** | $\mathbf{38.95 \pm 2.03}$ | $\mathbf{24.92 \pm 1.57}$ |

**Table 2:** Accuracy of Data Weighting on Image Classification. The small subset of CIFAR10 used here has 40 training instances and 2 validation instances for each class. The "pretrained" column is the results by initializing the ResNet-34 [14] base model with ImageNet-pretrained weights. In contrast, "Not-Pretrained" denotes the base model is randomly initialized. Since every class has the same number of examples, the proportion-based weighting degenerates to base model training and thus is omitted here.

with a factor of 0.1 and then adding the gradient. All experiments were implemented with PyTorch (pytorch.org) and were performed on a Linux machine with 4 GTX 1080Ti GPUs and 64GB RAM. All reported results are averaged over 15 runs $\pm$ one standard deviation.

## 5.2 Low Data Regime

We study the problem where only very few labeled examples for each class are available. Both of our augmentation and weighting boost base model performance, and are superior to respective comparison methods. We also observe that augmentation performs better than weighting in the low-data setting.

**Setup** For text classification, we use the popular benchmark datasets, including SST-5 for 5-class sentence sentiment [45], IMDB for binary movie review sentiment [31], and TREC for 6-class question types [30]. We subsample a small training set on each task by randomly picking 40 instances for each class. We further create small validation sets, i.e., 2 instances per class for SST-5, and 5 instances per class for IMDB and TREC, respectively. The reason we use slightly more validation examples on IMDB and TREC is that the model can easily achieve 100% validation accuracy if the validation sets are too small. Thus, the SST-5 task has 210 labeled examples in total, while IMDB has 90 labels and TREC has 270. Such extremely small datasets pose significant challenges for learning deep neural networks. Since the manipulation parameters are trained using the small validation sets, to avoid possible overfitting we restrict the training to small number (e.g., 5 or 10) of epochs. For image classification, we similarly create a small subset of the CIFAR10 data, which includes 40 instances per class for training, and 2 instances per class for validation.

| Model | 20 : 1000 | 50 : 1000 | 100 : 1000 |
|---|---|---|---|
| Base model: BERT [7] | $54.91 \pm 5.98$ | $67.73 \pm 9.20$ | $75.04 \pm 4.51$ |
| Base model + val-data | $52.58 \pm 4.58$ | $55.90 \pm 4.18$ | $68.21 \pm 5.28$ |
| Proportion | $57.42 \pm 7.91$ | $71.14 \pm 6.71$ | $76.14 \pm 5.8$ |
| Ren et al. [39] | $74.61 \pm 3.54$ | $76.89 \pm 5.07$ | $80.73 \pm 2.19$ |
| **Ours** | $\mathbf{75.08 \pm 4.98}$ | $\mathbf{79.35 \pm 2.59}$ | $\mathbf{81.82 \pm 1.88}$ |

**Table 3:** Accuracy of Data Weighting on Imbalanced SST-2. The first row shows the number of training examples in each of the two classes.

| Model | 20 : 1000 | 50 : 1000 | 100 : 1000 |
|---|---|---|---|
| Base model: ResNet [14] | $72.20 \pm 4.70$ | $81.65 \pm 2.93$ | $86.42 \pm 3.15$ |
| Base model + val-data | $64.66 \pm 4.81$ | $69.51 \pm 2.90$ | $79.38 \pm 2.92$ |
| Proportion | $72.29 \pm 5.67$ | $81.49 \pm 3.83$ | $84.26 \pm 4.58$ |
| Ren et al. [39] | $74.35 \pm 6.37$ | $82.25 \pm 2.08$ | $86.54 \pm 2.69$ |
| **Ours** | $\mathbf{75.32 \pm 6.36}$ | $\mathbf{83.11 \pm 2.08}$ | $\mathbf{86.99 \pm 3.47}$ |

**Table 4:** Accuracy of Data Weighting on Imbalanced CIFAR10. The first row shows the number of training examples in each of the two classes.

**Results** Table 1 shows the manipulation results on text classification. For data augmentation, our approach significantly improves over the base model on all the three datasets. Besides, compared to both the conventional synonym substitution and the approach that keeps the augmentation network fixed, our adaptive method that fine-tunes the augmentation network jointly with model training achieves superior results. Indeed, the heuristic-based synonym approach can sometimes harm the model performance (e.g., SST-5 and IMDB), as also observed in previous work [49, 25]. This can be because the heuristic rules do not fit the task or datasets well. In contrast, learning-based augmentation has the advantage of adaptively generating useful samples to improve model training.

Table 1 also shows the data weighting results. Our weight learning consistently improves over the base model and the latest weighting method [39]. In particular, instead of re-estimating sample weights from scratch in each iteration [39], our approach treats the weights as manipulation parameters maintained throughout the training. We speculate that the parametric treatment can adapt weights more smoothly and provide historical information, which is beneficial in the small-data context.

It is interesting to see from Table 1 that our augmentation method consistently outperforms the weighting method, showing that data augmentation can be a more suitable technique than data weighting for manipulating small-size data. Our approach provides the generality to instantiate diverse manipulation types and learn with the same single procedure.

To investigate the augmentation model and how the fine-tuning affects the augmentation results, we show in Figure 2 the top-5 most probable word substitutions predicted by the augmentation model for two masked tokens, respectively. Comparing the results of epoch 1 and epoch 3, we can see the augmentation model evolves and dynamically adjusts the augmentation behavior as the training proceeds. Through fine-tuning, the model seems to make substitutions that are more coherent with the conditioning label and relevant to the original words (e.g., replacing the word "striking" with "bland" in epoch 1 v.s. "charming" in epoch 3).

Table 2 shows the data weighting results on image classification. We evaluate two settings with the ResNet-34 base model being initialized randomly or with pretrained weights, respectively. Our data weighting consistently improves over the base model and [39] regardless of the initialization.

## 5.3 Imbalanced Labels

We next study a different problem setting where the training data of different classes are imbalanced. We show the data weighting approach greatly improves the classification performance. It is also observed that, the LM data augmentation approach, which performs well in the low-data setting, fails on the class-imbalance problems.

**Setup** Though the methods are broadly applicable to multi-way classification problems, here we only study binary classification tasks for simplicity. For text classification, we use the SST-2 sentiment analysis benchmark [45]; while for image, we select class 1 and 2 from CIFAR10 for binary classification. We use the same processing on both datasets to build the class-imbalance setting. Specifically, we randomly select 1,000 training instances of class 2, and vary the number of class-1 instances in $\{20, 50, 100\}$. For each dataset, we use 10 validation examples in each class. Trained models are evaluated on the full binary-class test set.

**Results** Table 3 shows the classification results on SST-2 with varying imbalance ratios. We can see our data weighting performs best across all settings. In particular, the improvement over the base model increases as the data gets more imbalanced, ranging from around 6 accuracy points on 100:1000 to over 20 accuracy points on 20:1000. Our method is again consistently better than [39], validating that the parametric treatment is beneficial. The proportion-based data weighting provides only limited improvement, showing the advantage of adaptive data weighting. The base model trained on the joint training-validation data for fixed steps fails to perform well, partly due to the lack of a proper mechanism for selecting steps.

Table 4 shows the results on imbalanced CIFAR10 classification. Similarly, our method outperforms other comparison approaches. In contrast, the fixed proportion-based method sometimes harms the performance as in the 50:1000 and 100:1000 settings.

We also tested the text augmentation LM on the SST-2 imbalanced data. Interestingly, the augmentation tends to hinder model training and yields accuracy of around 50% (random guess). This is because the augmentation LM is first fit to the imbalanced data, which makes label preservation inaccurate and introduces lots of noise during augmentation. Though a more carefully designed augmentation mechanism can potentially help with imbalanced classification (e.g., augmenting only the rare classes), the above observation further shows that the varying data manipulation schemes have different applicable scopes. Our approach is thus favorable as the single algorithm can be instantiated to learn different schemes.

## 6    Discussions: Algorithm Extrapolation between Learning Paradigms

**Conclusions.** We have developed a new method of learning different data manipulation schemes with the same single algorithm. Different manipulation schemes reduce to just different parameterization of the data reward function. The manipulation parameters are trained jointly with the target model parameters. We instantiate the algorithm for data augmentation and weighting, and show improved performance over strong base models and previous manipulation methods. We are excited to explore more types of manipulations such as data synthesis, and in particular study the combination of different manipulation schemes.

The proposed method builds upon the connections between supervised learning and reinforcement learning (RL) [46] through which we extrapolate an off-the-shelf reward learning algorithm in the RL literature to the supervised setting. The way we obtained the manipulation algorithm represents a general means of innovating problem solutions based on unifying formalisms of different learning paradigms. Specifically, a unifying formalism not only offers new understandings of the seemingly distinct paradigms, but also allows us to systematically apply solutions to problems in one paradigm to similar problems in another. Previous work along this line has made fruitful results in other domains. For example, an extended formulation of [46] that connects RL and posterior regularization (PR) [11, 16] has enabled to similarly export a reward learning algorithm to the context of PR for learning structured knowledge [18]. By establishing a uniform abstration of GANs [13] and VAEs [23], Hu et al. [19] exchange techniques between the two families and get improved generative modeling. Other work in the similar spirit includes [40, 41, 9, etc].

By extrapolating algorithms between paradigms, one can go beyond crafting new algorithms from scratch as in most existing studies, which often requires deep expertise and yields unique solutions in a dedicated context. Instead, innovation becomes easier by importing rich ideas from other paradigms, and is repeatable as a new algorithm can be methodically extrapolated to multiple different contexts.

## Footnotes

[1]Code available at https://github.com/tanyuqian/learning-data-manipulation

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
