[Reviews · NeurIPS 2019]

Reviewer 1



I think the paper was very insightful and represents a significant advance. It was well written and well explained. It combines two powerful ideas, using feedback from a small validation set, and a fresh theoretical perspective linking RL ideas to the gradient descent realm. ------------------------------------------- Post Rebuttal --------------------------------------- I've read the author rebuttal and other reviews. The authors seems to have replied to issues other reviewers had in appreciating the novelty/significance. Myself, I still honestly think that if I randomly sampled 19 other accepted NIPS papers and this one, it would be the most significant and interesting read. I'm comfortable remaining an isolated outlier for this evaluation.

Reviewer 2



The paper propose a system that takes advantage of combining an off-the-shelf RL model with supervised training to create different parameterizations of "data reward", which helps guide the weighting and augmentation of training data. The method is clearly described; however, given the idea of RL for data weighting/augmentation has been explored before, it is challenging to judge what the specific novelty is.= beyond the new data reward *function* itself. The experiments are thorough and support the advantages of the algorithm. Table 1 and Table 2 show the performance of the method in the augmentation and the weighting conditions, but not when *both* are applied simultaneously. Next, the method is not tested for augmentation over the class imbalanced CIFAR dataset. The main drawback of the method is that the training data cannot be weighted and augmented simultaneously. Given the algorithm is general enough and based on the same data reward function, combining these two popular data manipulation strategies would be interesting.

Reviewer 3



Originality: The proposed framework is fairly novel and provides an interesting perspective on learning data manipulation. I found the Quality: The experiments on the text classification show that the proposed algorithms work well. However I found the experiments on image classification setting to be not very convincing~(see Improvements section) Clarity: The paper is well written and organized and contains sufficient details to enable reproducing the results. Significance: The proposed algorithm is flexible to incorporate different data manipulation schemes and provides a method to learn them to improve the end-task. This might enable integrating data generation methods~(GANs, VAEs) and learning an effective task-specific data-augmentation algorithms. ------------------------------------------- Post Rebuttal --------------------------------------- Having read the other reviews and the authors rebuttal, I stick to my initial evaluation that this is a good paper and should be accepted. The rebuttal provides satisfactory answers to my questions.

[Author Response · NeurIPS 2019]

We thank the reviewers for their valuable and encouraging feedback.

**Reviewer #1:**

Thank you for the supportive comments!

It is straightforward to adapt the AutoAugment policy model into our framework, by simply parameterizing the
augmentation model $g_\phi(x|x^*, y)$ in Eq.(9) as the policy used in AutoAugment. If the policy contains discrete
components to be learned (i.e., $\phi$ has discrete factors), we can use policy gradient for optimizing $\phi$.

The present work has primarily focused on the generality of the proposed framework. We thus tested in both text and
image domains, with data augmentation and weighting. We are excited to apply the approach in more problem settings
and manipulation schemes, and compare with other work including AutoAugment in the future.

We will polish the writing as pointed out. Thank you for the suggestion!

**Reviewer #2:**

* *Novelty:*

The core novelty of the approach is the generality of the formulation, in which different manipulation schemes boil
down to different parameterization of the data reward function. The generality enables us to study augmentation
and weighting in text and image domains, which differs from previous work that typically applies to a single type of
manipulation and often in a single domain.

The resulting augmentation/manipulation algorithms also differ from previous RL-based methods, as explained in
Line.151–156. In particular, learning manipulation in our algorithms is carried out by learning the *reward* function,
which is a new perspective compared to previous work that learns a *policy* [e.g., 4]. The (intrinsic) reward learning
procedure we adopted enables efficient iterative optimization of the model and the manipulation. We will summarize
the novelty clearer in the revised version.

* *Simultaneous weighting and augmentation*

The primary focus of the present work is to develop the general framework that supports a variety of manipulation
schemes. We have studied the effectiveness of the approach in richer settings than previous work, including weighting
and augmentation on text and/or images. As pointed out by the reviewer, our approach can also naturally enable
simultaneous weighting and augmentation (by parameterizing the data reward function accordingly). We apply the
simultaneous manipulation on the imbalanced SST-2 task (Table 3), where we only augment the rare class, and induce
weights for both the real and augmented data. In 50:1000 and 100:1000 settings, we achieve $81.62 \pm 2.26$ and
$82.39 \pm 2.04$ accuracy, respectively, which improve over the weighting-only results by around 1–1.5 accuracy points
(Table 3). We will provide more complete results in the revised version.

* *Augmentation over CIFAR data*

We tested data augmentation in the text domain which is less well-studied than image augmentation. Our approach can
support augmenting images by parameterizing the augmentation model $g_\phi(x|x^*, y)$ (Eq.9) as an image augmentation
model. We leave the study in future work.

**Reviewer #3:**

* *Data weighting in low data regime*

Data weighting helps low-data tasks by emphasizing important data points so that the small datasets are used in a more
effective way. The results in Table 2 verify the effectiveness. Besides, comparing data weighting and augmentation
in the low-data regime shows different effectiveness of the two manipulation schemes (Table 1, augmentation v.s.
weighting), which highlights the need of a general approach that enables different manipulation through simple variation
(in the data reward function). We agree that the noisy-label task, similar to the class-imbalance setting which we have
studied, is another great application for data weighting. We expect to study this setting in the future.

* *Class-imbalance data*

We have followed the experiment setup in Ren et al. [26] which also studied on a two-class imbalanced (MNIST) data.
Our approach has shown consistent improvement over Ren et al. [26] in both text and image imbalance settings. Also
note that the low-data tasks in Table 1 (SST-2 and TREC) are multi-class settings. Applying the approach in more
contexts including naturally imbalanced datasets is an exacting direction to investigate in future work.

[Meta-Review · NeurIPS 2019]

The paper presents a gradient based meta learning approach to automating data augmentation and weighting examples. The experiments support the advantages of the proposed technique. There has been recent work on using gradient based meta-learning for weighting examples (Ren et al., ICML, 2018 [26]) and learning reward functions (Agarwal et al., ICML 2019 [https://arxiv.org/pdf/1902.07198.pdf]). There are some interesting technical novelty in the proposed algorithm and a clear discussion of such novelty in the context of recent papers is beneficial. Given the similarity of the proposed technique and existing recent work on meta-learning, I recommend accepting as a poster.